# Fast Distributed Submodular Cover:
# Public-Private Data Summarization

**Baharan Mirzasoleiman**
ETH Zurich

**Morteza Zadimoghaddam**
Google Research

**Amin Karbasi**
Yale University

## Abstract

In this paper, we introduce the public-private framework of data summarization motivated by privacy concerns in personalized recommender systems and online social services. Such systems have usually access to massive data generated by a large pool of users. A major fraction of the data is public and is visible to (and can be used for) all users. However, each user can also contribute some private data that should not be shared with other users to ensure her privacy. The goal is to provide a succinct summary of massive dataset, ideally as small as possible, from which customized summaries can be built for each user, i.e. it can contain elements from the public data (for diversity) and users' private data (for personalization).

To formalize the above challenge, we assume that the scoring function according to which a user evaluates the utility of her summary satisfies submodularity, a widely used notion in data summarization applications. Thus, we model the data summarization targeted to each user as an instance of a submodular cover problem. However, when the data is massive it is infeasible to use the centralized greedy algorithm to find a customized summary even for a single user. Moreover, for a large pool of users, it is too time consuming to find such summaries separately. Instead, we develop a fast distributed algorithm for submodular cover, FASTCOVER, that provides a succinct summary in *one shot* and for *all users*. We show that the solution provided by FASTCOVER is competitive with that of the centralized algorithm with the number of rounds that is exponentially smaller than state of the art results. Moreover, we have implemented FASTCOVER with Spark to demonstrate its practical performance on a number of concrete applications, including personalized location recommendation, personalized movie recommendation, and dominating set on tens of millions of data points and varying number of users.

## 1 Introduction

Data summarization, a central challenge in machine learning, is the task of finding a representative subset of manageable size out of a large dataset. It has found numerous applications, including image summarization [1], recommender systems [2], scene summarization [3], clustering [4, 5], active set selection in non-parametric learning [6], and document and corpus summarization [7, 8], to name a few. A general recipe to obtain a faithful summary is to define a utility/scoring function that measures coverage and diversity of the selected subset [1]. In many applications, the choice of utility functions used for summarization exhibit submodularity, a natural diminishing returns property. In words, submodularity implies that the added value of any given element from the dataset decreases as we include more data points to the summary. Thus, the data summarization problem can be naturally reduced to that of a submodular cover problem where the objective is to find the *smallest* subset whose utility achieves a desired fraction of the utility provided by the entire dataset.

It is known that the classical greedy algorithm yields a logarithmic factor approximation to the optimum summary [9]. It starts with an empty set, and at each iteration adds an element with the

maximum added value to the summary selected so far. It is also known that improving upon the logarithmic approximation ratio is NP-hard [10]. Even though the greedy algorithm produces a near-optimal solution, it is highly impractical for massive datasets, as sequentially selecting elements on a single machine is heavily constrained in terms of speed and memory. Hence, in order to solve the submodular cover problem at scale, we need to make use of MapReduce-style parallel computation models [11, 12]. The greedy algorithm, due to its sequential nature, is poorly suited for parallelization.

In this paper, we propose a fast distributed algorithm, FASTCOVER, that enables us to solve the more general problem of covering multiple submodular functions in one run of the algorithm. It relies one three important ingredients: 1) a reduction from multiple submodular cover problems into a single instance of a submodular cover problem [13, 14], 2) randomized filtration mechanism to select elements with high utility, and 3) a set of carefully chosen threshold functions used for the filteration mechanism. FASTCOVER also provides a natural tarde-off between the number of MapReduce rounds and the size of the returned summary. It effectively lets us choose between compact summaries (i.e., smaller solution size) while running more MapReduce rounds or larger summaries while running fewer MapReduce rounds.

This setting is motivated by privacy concerns in many modern applications, including personalized recommender systems, online social services, and the data collected by apps on mobile platforms [15, 16]. In such applications, users have some control over their own data and can mark some part of it private (in a slightly more general case, we can assume that users can make part of their data private to specific groups and public to others). As a result, the dataset consists of public data, shared among all users, and disjoint sets of private data accessible to the owners only.

We call this more general framework for data summarization, public-private data summarization, where the private data of one user should not be included in another user's summary (see also [15]). This model naturally reduces to solving one instance of the submodular cover problem for each user, as their view of the dataset and the specific utility function specifying users' preferences differ across users. When the number of users is small, one can solve the public-private data summarization separately for each user, using the greedy algorithm (for datasets of small size) or the recently proposed distributed algorithm DISCOVER [12] (for datasets of moderate size). However, when there are many users or the dataset is massive, none of the prior work truly scales.

We report performance of DISCOVER using Spark on concrete applications of the public-private data summarization, including personalized movie recommendation on a dataset containing 2 million ratings by more than 100K users for 1000 movies, personalized location recommendation based on 20 users and their collected GPS locations, and finding the dominating set on a social network containing more than 65 million nodes and 1.8 billion edges. For small to moderate sized datasets, we compare our results with previous work, namely, classical greedy algorithm and DISCOVER [12]. For truly large-scale experiments, where the data is big and/or there are many users involved (e.g., movie recommendation), we cannot run DISCOVER as the number of MapReduce rounds in addition to their communication costs is prohibitive. In our experiments, we constantly observe that FASTCOVER provides solutions of size similar to the greedy algorithm (and very often even smaller) with the number of rounds that are orders of magnitude smaller than DISCOVER. This makes FASTCOVER the first distributed algorithm that solves the public-private data summarization fast and at scale.

## 2   Problem Statement: Public-Private Data Summarization

In this section, we formally define the public-private model of data summarization[1]. Here, we consider a potentially large dataset (sometimes called universe of items) $\mathbb{V}$ of size $n$ and a set of users $\mathbb{U}$. The dataset consists of public data $\mathbb{V}^P$ and disjoint subsets of private data $\mathbb{V}_u$ for each user $u \in \mathbb{U}$. The public-private aspect of data summarization realizes in two dimensions. First, each user $u \in \mathbb{U}$ has her own utility function $f_u(S)$ according to which she scores the value of a subset $S \subseteq \mathbb{V}$. Throughout this paper we assume that $f_u(\cdot)$ is integer-valued[2], non-negative, and monotone

submodular. More formally, submodularity means that

$$f_u(A \cup \{e\}) - f_u(A) \geq f_u(B \cup \{e\}) - f_u(B) \quad \forall A \subseteq B \subset \mathbb{V} \text{ and } \forall e \in \mathbb{V} \setminus B.$$

Monotonicity implies that for any $A \subseteq \mathbb{V}$ and $e \in \mathbb{V}$ we have $\Delta_{f_u}(e|A) \doteq f_u(A \cup \{e\}) - f_u(A) \geq 0$. The term $\Delta_{f_u}(e|A)$ is called the marginal gain (or added value) of $e$ to the set $A$. Whenever it is clear from the context we drop $f_u$ from $\Delta_{f_u}(e|A)$. Without loss of generality, we normalize all users' functions so that they achieve the same maximum value, i.e., $f_u(\mathbb{V}) = f_v(\mathbb{V})$ for all $u, v \in \mathbb{U}$. Second, and in contrast to public data that is shared among all users, the private data of a user cannot be shared with others. Thus, a user $u \in \mathbb{U}$ can only evaluate the public and her own private part of a summary $S$, i.e., $S \cap (\mathbb{V}^P \cup \mathbb{V}_u)$. In other words, if the summary $S$ contains private data of a user $v \neq u$, the user $u$ cannot have access or evaluate $v$'s private part of $S$, i.e., $S \cap \mathbb{V}_v$. In public-private data summarization, we would like to find the smallest subset $S \subseteq \mathbb{V}$ such that all users reach a desired utility $Q \leq f_u(\mathbb{V}) = f_u(\mathbb{V}^P \cup \mathbb{V}_u)$ simultaneously, i.e.,

$$\text{OPT} = \arg\min_{S \subseteq \mathbb{V}} |S|, \text{ such that } f_u(S \cap (\mathbb{V}^P \cup \mathbb{V}_u)) \geq Q \; \forall u \in \mathbb{U}. \tag{1}$$

A naive way to solve the above problem is to find a separate summary for each user and then return the union of all summaries as $S$. A more clever way is to realize that problem (1) is in fact equivalent to the following problem [13, 14]

$$\text{OPT} = \arg\min_{S \subseteq \mathbb{V}} |S|, \text{ such that } f(S) \doteq \sum_{u \in \mathbb{U}} \min\{f_u(S \cap (\mathbb{V}^P \cup \mathbb{V}_u)), Q\} \geq Q \times |\mathbb{U}|. \tag{2}$$

Note that the surrogate function $f(\cdot)$ is also monotone submodular as a thresholded submodular function remains submodular. Thus, finding a set $S$ that provides each user with utility $Q$ is equivalent of finding a set $S$ with $f(S) \geq L \doteq Q \times |\mathbb{U}|$. This reduction lets us focus on developing a fast distributed solution for solving a single submodular cover problem. Our method FASTCOVER is explained in detail in Section 4.

**Related Work:** When the data is small, we can use the centralized greedy algorithm to solve problem (2) (and equivalently problem (1)). The greedy algorithm sequentially picks elements and returns a solution of size $(1 + \ln M)\text{OPT} \approx \ln(L)|\text{OPT}|$ where $M = \max_{e \in \mathbb{V}} f(e)$. As elaborated earlier, when the data is large, one cannot run this greedy algorithm as it requires centralized access to the full dataset. This is why scalable solutions for the submodular cover problem have recently gained a lot of interest. In particular, for the set cover problem (a special case of submodular cover problem) there have been efficient MapReduce-based implementations proposed in the literature [17, 18, 19]. There have also been recent studies on the streaming set cover problem [20]. Perhaps the closest work to our efforts is [12] where the authors proposed a distributed algorithm for the submodular cover problem called DISCOVER. Their method relies on the reduction of the submodular cover problem to multiple instances of the distributed constrained submodular maximization problem [6, 21]. For any fixed $0 < \alpha \leq 1$, DISCOVER returns a solution of size $\lceil 2\alpha k + 72 \log(L)|\text{OPT}|\sqrt{\min(m, \alpha|\text{OPT}|)}) \rceil$ in $\lceil \log(\alpha|\text{OPT}|) + 36\sqrt{\min(m, \alpha|\text{OPT}|)} \log(L)/\alpha + 1 \rceil$ rounds, where $m$ denotes the number of machines. Even though DISCOVER scales better than the greedy algorithm, the solution it returns is usually much larger. Moreover, the dependency of the number of MapReduce rounds on $\sqrt{\min(m, \alpha|\text{OPT}|)}$ is far from desirable. Note that as we increase the number of machines, the number of rounds may increase (rather than decreasing). Instead, in this paper we propose a fast distributed algorithm, FASTCOVER, that truly scales to massive data and produces a solution that is competitive with that of the greedy algorithm. More specifically, for any $\epsilon > 0$, FASTCOVER returns a solution of size at most $\lceil \ln(L)|\text{OPT}|/(1-\epsilon) \rceil$ with at most $\lceil \log_{3/2}(n/m|\text{OPT}|) \log(M)/\epsilon + \log(L) \rceil$ rounds, where $M = \max_{e \in \mathbb{V}} f(e)$. Thus, in terms of speed, FASTCOVER improves exponentially upon DISCOVER while providing a smaller solution. Moreover, in our work, the number of rounds decreases as the number of machines increases, in sharp contrast to [12].

## 3 Applications of Pubic-Private Data Data Summarization

In this section, we discuss 3 concrete applications where parts of data are private and the remaining parts are public. All objective functions are non-negative, monotone, and submodular.

## 3.1 Personalized Movie Recommendation

Consider a movie recommender system that allows users to anonymously and privately rate movies. The system can use this information to recognize users' preferences using existing matrix completion techniques [22]. A good set of recommended movies should meet two criteria: 1) be correlated with user's preferences, and 2) be diverse and contains globally popular movies. To this end, we define the following sum-coverage function to score the quality of the selected movies $S$ for a user $u$:

$$f_u(S) = \alpha_u \sum_{i \in S, j \in \mathbb{V}_u} s_{i,j} + (1 - \alpha_u) \sum_{i \in S, j \in \mathbb{V}^P \setminus S} s_{i,j}, \qquad (3)$$

where $\mathbb{V}_u$ is the list of highly ranked movies by user $u$ (i.e., private information), $\mathbb{V}^P$ is the set of all movies in the database[3], and $s_{i,j}$ measures the similarity between movie $i$ and $j$. The similarity can be easily calculated using the inner product between the corresponding feature vectors of any two movies $i$ and $j$. The term $\sum_{i \in S, j \in \mathbb{V}_u} s_{i,j}$ measures the similarity between the recommended set $S$ and the user's preferences. The second term $\sum_{i \in S, j \in \mathbb{V}^P \setminus S} s_{i,j}$ encourages diversity. Finally, the parameter $0 \leq \alpha_u \leq 1$ provides the user the freedom to specify how much she cares about personalization versus diversity, i.e., $\alpha_u = 1$ indicates that all the recommended movies should be very similar to the movies she highly ranked and $\alpha_u = 0$ means that she prefers to receive a set of globally popular movies among all users, irrespective of her own private ratings. Note that in this application, the universe of items (i.e., movies) is public. What is private is the users' ratings through which we identify the set of highly ranked movies by each user $\mathbb{V}_u$. The effect of private data is expressed in users' utility functions. The objective is to find the smallest set $S$ of movies $\mathbb{V}$, from which we can build recommendations for all users in a way that all reach a certain utility.

## 3.2 Personalized Location Recommendation

Nowadays, many mobile apps collect geolocation data of their users. To comply with privacy concerns, some let their customers have control over their data, i.e., users can mark some part of their data private and disallow the app to share it with other users. In the personalized location recommendation, a user is interested in identifying a set of locations that are correlated with the places she visited and popular places everyone else visited. Note that as close by locations are likely to be similar it is very typical to define a kernel matrix $\mathcal{K}$ capturing the similarity between data points. A commonly used kernel in practice is the squared exponential kernel $\mathcal{K}(e_i, e_j) = \exp(-||e_i - e_j||_2^2/h^2)$. To define the information gain of a set of locations indexed by $S$, it is natural to use $f(S) = \log \det(I + \sigma \mathcal{K}_{S,S})$. The information gain objective captures the diversity and is used in many ML applications, e.g., active set selection for nonparametric learning [6], sensor placement [13], determinantal point processes, among many others. Then, the personalized location recommendation can be modeled by

$$f_u(S) = \alpha_u f(S \cap \mathbb{V}_u) + (1 - \alpha_u) f(S \cap \mathbb{V}^P), \qquad (4)$$

where $\mathbb{V}_u$ is the set of locations that user $u$ does not want to share with others and $\mathbb{V}^P$ is the collection of all publicly disclosed locations. Again, the parameter $\alpha_u$ lets the user indicate to what extent she is willing to receive recommendations based on her private information. The objective is to find the smallest set of locations to recommend to all users such that each reaches a desired threshold. Note that private data is usually small and private functions are fast to compute. Thus, the function evaluation is mainly affected by the amount of public data. Moreover, for many objectives, e.g., information gain, each machine can evaluate $f_u(S)$ by using its own portion of the private data.

## 3.3 Dominating Set in Social Networks

Probably the easiest way to define the influence of a subset of users on other members of a social network is by the dominating set problem. Here, we assume that there is a graph $G = (\mathbb{V}, E)$ where $\mathbb{V}$ and $E$ indicate the set of nodes and edges, respectively. Let $\mathcal{N}(S)$ denote the neighbors of $S$. Then, we define the coverage size of $S$ by $f(S) = |\mathcal{N}(S) \cup S|$. The goal is to find the smallest subset $S$ such that the coverage size is at least some fraction of $|\mathbb{V}|$. This is a trivial instance of public-private data summarization as all the data is public and there is a single utility function. We use the dominating set problem to run a large-scale application for which DISCOVER terminates in a reasonable amount of time and its performance can be compared to our algorithm FASTCOVER.

# 4 FASTCOVER for Fast Distributed Submodular Cover

In this section, we explain in detail our fast distributed Algorithm FASTCOVER shown in Alg. 1. It receives a universe of items $\mathbb{V}$ and an integer-valued, non-negative, monotone submodular function $f : 2^{\mathbb{V}} \rightarrow \mathbb{R}_+$. The objective is to find the smallest set $S$ that achieves a value $L \leq f(\mathbb{V})$. FASTCOVER starts with $S = \emptyset$, and keeps adding those items $x \in \mathbb{V}$ to $S$ whose marginal values $\Delta(e|S)$ are at least some threshold $\tau$. In the beginning, $\tau$ is set to a conservative initial value $M \doteq \max_{x \in \mathbb{V}} f(x)$. When there are no more items with a marginal value $\tau$, FASTCOVER lowers $\tau$ by a factor of $(1 - \epsilon)$, and iterates anew through the elements. Thus, $\tau$ ranges over $\tau_0 = M, \tau_1 = (1 - \epsilon)M, \cdots, \tau_\ell = (1 - \epsilon)^\ell M, \cdots$. FASTCOVER terminates when $f(S) \geq L$. The parameter $\epsilon$ determines the size of the final solution. When $\epsilon$ is small, we expect to find better solutions (i.e., smaller in size) while having to spend more number of rounds.

One of the key ideas behind FASTCOVER is that finding elements with marginal values $\tau = \tau_\ell$ can be done in a distributed manner. Effectively, FASTCOVER partitions $\mathbb{V}$ into $m$ sets $T_1, \ldots, T_m$, one for each cluster node/machine. A naive distributed implementation is the following. For a given set $S$ (whose elements are communicated to all machines) each machine $i$ finds all of its items $x \in T_i$ whose marginal values $\Delta(x|S)$ are larger than $\tau$ and send them all to a central machine (note that $S$ is fixed on each machine). Then, this central machine sequentially augments $S$ with elements whose marginal values are more than $\tau$ (here $S$ changes by each insertion). The new elements of $S$ are communicated back to all machines and they run the same procedure, this time with a smaller threshold $\tau(1 - \epsilon)$. The main problem with this approach is that there might be many items on each machine that satisfy the chosen threshold $\tau$ at each round (i.e., many more than |OPT|). A flood of such items from $m$ machines overwhelms the central machine. Instead, what FASTCOVER does is to enforce each machine to randomly pick only $k$ items from their potentially big set of candidates (i.e., THRESHOLDSAMPLE algorithm shown in Alg. 2). The value $k$ is carefully chosen (line 7). This way the number of items the central machine processes is never more than $O(m|\text{OPT}|)$.

| |
|---|
| **1** **Input:** $\mathbb{V}$, $\epsilon$, $L$, and $m$ |
| **2** **Output:** $S \subseteq \mathbb{V}$ where $f(S) \geq L$ |
| **3** Find a balanced partition $\{T_i\}_{i=1}^m$ of $\mathbb{V}$; |
| **4** $S \leftarrow \emptyset$; |
| **5** $\tau \leftarrow \max_{x \in \mathbb{V}} f(x)$; |
| **6** **while** $\tau \geq 1$ **do** |
| **7** $\quad k \leftarrow \lceil (L - f(S))/\tau \rceil$; |
| **8** $\quad$ **forall the** $1 \leq i \leq m$ **do** |
| **9** $\quad\quad <S_i, Full_i> \leftarrow ThresholdSample(i, \tau, k, S)$; |
| **10** $\quad$ **forall the** $x \in \cup_{i=1}^m S_i$ **do** |
| **11** $\quad\quad$ **if** $f(\{x\} \cup S) - f(S) \geq \tau$ **then** |
| **12** $\quad\quad\quad S \leftarrow S \cup \{x\}$; |
| **13** $\quad\quad\quad$ **if** $f(S) \geq L$ **then** Break; |
| **14** $\quad$ **if** $\forall i : Full_i = False$ **then** |
| **15** $\quad\quad$ **if** $\tau > 1$ **then** $\tau \leftarrow \max\{1, (1 - \epsilon)\tau\}$; |
| **16** $\quad\quad$ **else** Break; |
| **17** Return $S$; |
| **Algorithm 1:** FASTCOVER |

| |
|---|
| **1** **Input:** Index $i$, $\tau$, $k$, and $S$ |
| **2** **Output:** $S_i \subset T_i$ with $|S_i| \leq k$ |
| **3** $S_i \leftarrow \emptyset$; |
| **4** **forall the** $x \in S_i$ **do** |
| **5** $\quad$ **if** $f(S \cup \{x\}) - f(S) \geq \tau$ **then** |
| **6** $\quad\quad S_i \leftarrow S_i \cup \{x\}$; |
| **7** **if** $|S_i| \leq k$ **then** |
| **8** $\quad$ Return $< S_i, False >$; |
| **9** **else** |
| **10** $\quad S_i \leftarrow k$ random items of $S_i$; |
| **11** $\quad$ Return $< S_i, True >$; |
| **Algorithm 2:** THRESHOLDSAMPLE |

**Theorem 4.1.** FASTCOVER *terminates with at most* $\log_{3/2}(n/(|\text{OPT}|m))(1+\log(M)/\epsilon)+\log_2(L)$ *rounds (with high probability) and a solution of size at most* $|\text{OPT}|\ln(L)/(1 - \epsilon)$.

Although FASTCOVER is distributed and unlike centralized algorithms does not enjoy the benefits of accessing all items together, its solution size is truly competitive with the greedy algorithm and is only away by a factor of $1/(1 - \epsilon)$. Moreover, its number of rounds is logarithmic in $n$ and $L$. This is in sharp contrast with the previously best known algorithm, DISCOVER [12], where the number of rounds scales with $\sqrt{\min(m, |OPT|)}$[4]. Thus, FASTCOVER not only improves exponentially over

DISCOVER in terms of speed but also its number of rounds decreases as the number of available machines $m$ increases. Even though FASTCOVER is a simple distributed algorithm, its performance analysis is technical and is deferred to the supplementary materials. Below, we provide the main ideas behind the proof of Theorem 4.1.

**Proof sketch.** *We say that an item has a high value if its marginal value to $S$ is at least $\tau$. We define an epoch to be the rounds during which $\tau$ does not change. In the last round of each epoch, all high value items are sent to the central machine (i.e., the set $\cup_{i=1}^{m}S_i$) because $Full_i$ is false for all machines. We also add every high value item to $S$ in lines $11-12$. So, at the end of each epoch, marginal values of all items to $S$ are less than $\tau$. Since we reduce $\tau$ by a factor of $(1-\epsilon)$, we can always say that $\tau \geq (1-\epsilon)\max_{x\in\mathbb{V}}\Delta(x|S)$ which means we are only adding items that have almost the highest marginal values. By the classic analysis of greedy algorithm for submodular maximization, we can conclude that every item we add has an added value that is at least $(1-\epsilon)(L-f(S))/|\text{OPT}|$. Therefore, after adding $|\text{OPT}|\ln(L)/(1-\epsilon)$ items, $f(S)$ becomes at least $L$.*

*To upper bound rounds, we divide the rounds into two groups. In a good round, the algorithm adds at least $\frac{k}{2}$ items to $S$. The rest are bad rounds. In a good round, we add $k/2 \geq (L-f(S))/(2\tau)$ items, and each of them increases the value of $S$ by $\tau$. Therefore in a good round, we see at least $(L-f(S))/2$ increase in value of $S$. In other words, the gap $L - f(S)$ is reduced by a factor of at least 2 in each good round. Since $f$ only takes integer values, once $L - f(S)$ becomes less than 1, we know that $f(S) \geq L$. Therefore, there cannot be more than $\log_2 L$ good rounds. Every time we update $\tau$ (start of an epoch), we decrease it by a factor of $1 - \epsilon$ (except maybe the last round for which $\tau = 1$). Therefore, there are at most $1 + \log_{\frac{1}{1-\epsilon}}(M) \leq 1 + \frac{\log(M)}{\log(1/(1-\epsilon))} \leq 1 + \frac{\log(M)}{\epsilon}$ epochs. In a bad round, a machine with more than $k$ high value items, sends $k$ of those to the central machine, and at most $k/2$ of them are selected. In other words, the addition of these items to $S$ in this bad round caused more than half of high value items of each machine to become of low value (marginal values less than $\tau$). Since there are $n/m$ items in each machine, and $Full_i$ becomes False once there are at most $k$ high value items in the machine, we conclude that in expectation there should not be more than $\log_2(n/km)$ bad rounds in each epoch. Summarizing the upper bounds yields the bound on total number of rounds. Finer analysis leads to the high probability claim.* ☐

## 5 Experiments

In this section, we evaluate the performance of FASTCOVER on the three applications that we described in Section 3: personalized movie recommendation, personalized location recommendation, and dominating set on social networks. To validate our theoretical results and demonstrate the effectiveness of FASTCOVER, we compare the performance of our algorithm against DISCOVER and the centralized greedy algorithm (when possible).

Our experimental infrastructure was a cluster of 16 quad-core machines with 20GB of memory each, running Spark. The cluster was configured with one master node responsible for resource management, and the remaining 15 machines working as executors. We set the number of reducers to $m = 60$. To run FASTCOVER on Spark, we first distributed the data uniformly at random to the machines, and performed a map/reduce task to find the highest marginal gain $\tau = M$. Each machine then carries out a set of map/reduce tasks in sequence, where each map/reduce stage filters out elements with a specific threshold $\tau$ on the whole dataset. We then tune the parameter $\tau$, communicate back the results to the machines and perform another round of map/reduce calculation. We continue performing map/reduce tasks until we get to the desired value $L$.

### 5.1 Personalized Location Recommendation with Spark

Our location recommendation experiment involves applying FASTCOVER to the information gain utility function, described in Eq. (4). Our dataset consists of 3,056 GPS measurements from 20 users in the form of (latitude, longitude, altitude) collected during bike tours around Zurich [23]. The size of each path is between 50 and 500 GPS coordinates. For each pairs of points $i$ and $j$ we used the corresponding GPS coordinates to calculate their distance in meters $d(i,j)$ and then formed a squared exponential kernel $\mathcal{K}_{i,j} = \exp(-d(i,j)^2/h^2)$ with $h = 1500$. For each user, we marked 20% of her data private (data points are chosen consecutively) selected from each path taken by the biker. The parameter $\alpha_u$ is set randomly for each user $u$.

Figures 1a, 1b, 1c compare the performance of FASTCOVER to the benchmarks for building a recommendation set that covers 60%, 80%, and 90% of the maximum utility of each user. We

considered running DISCOVER with different values of parameter $\alpha$ that makes a trade off between the size of the solution and number of rounds of the algorithm. It can be seen that by avoiding the doubling steps of DISCOVER, our algorithm FASTCOVER is able to return a significantly smaller solution than that of DISCOVER in considerably less number of rounds. Interestingly, for small values of $\epsilon$, FASTCOVER returns a solution that is even smaller than the centralized greedy algorithm.

## 5.2 Personalized Movie Recommendation with Spark

Our personalized public-private recommendation experiment involves FASTCOVER applied to a set of 1,313 movies, and 20,000,263 users' ratings from 138,493 users of the MovieLens database [24]. All selected users rated at least 20 movies. Each movie is associated with a 25 dimensional feature vector calculated from users' ratings. We use the inner product of the non-normalized feature vectors to compute the similarity $s_{i,j}$ between movies $i$ and $j$ [25]. Our final objective function consists of 138,493 coverage functions -one per user- and a global sum-coverage function defined on the whole pool of movies (see Eq. (3)). Each function is normalized by its maximum value to make sure that all functions have the same scale.

Fig 1d, 1e, 1f show the ratio of the size of the solutions obtained by FASTCOVER to that of the greedy algorithm. The figures demonstrate the results for 10%, 20%, and 30% covers for all the 138,493 users' utility functions. The parameter $\alpha_u$ is set to 0.7 for all users. We scaled down the number of iterations by a factor of 0.01, so that the corresponding bars can be shown in the same figures. Again, FASTCOVER was able to find a considerably smaller solution than the centralized greedy. Here, we couldn't run DISCOVER because of its prohibitive running time on Spark.

Fig 1g shows the size of the solution set obtained by FASTCOVER for building recommendations from a set of 1000 movies for 1000 users vs. the size of the merged solutions found by finding recommendations separately for each user. It can be seen that FASTCOVER was able to find a much smaller solution by covering all the functions at the same time.

## 5.3 Large Scale Dominating Set with Spark

In order to be able to compare the performance of our algorithm with DISCOVER more precisely, we applied FASTCOVER to the Friendster network consists of 65,608,366 nodes and 1,806,067,135 edges [26]. This dataset was used in [12] to evaluate the performance of DISCOVER.

Fig. 1j, 1k, 1l show the performance of FASTCOVER for obtaining covers for 50%, 40%, 30% of the whole graph, compared to the centralized greedy solution. Again, the size of the solution obtained by FASTCOVER is smaller than the greedy algorithm for small values of $\epsilon$. Note that running the centralized greedy is impractical if the dataset cannot fit into the memory of a single machine. Fig. 1h compares the solution set size and the number of rounds for FASTCOVER and DISCOVER with different values of $\epsilon$ and $\alpha$. The points in the bottom left correspond to the solution obtained by FASTCOVER which confirm its superior performance. We further measured the actual running time of both algorithms on a smaller instance of the same graph with 14,043,721 nodes. We tuned $\epsilon$ and $\alpha$ to get solutions of approximately equal size for both algorithms. Fig. 1i shows the speedup of FASTCOVER over DISCOVER. It can be observed that by increasing the coverage value $L$, FASTCOVER shows an exponential speedup over DISCOVER.

## 6 Conclusion

In this paper, we introduced the public-private model of data summarization motivated by privacy concerns of recommender systems. We also developed a fast distributed algorithm, FASTCOVER, that provides a succinct summary for all users without violating their privacy. We showed that FASTCOVER returns a solution that is competitive to that of the best centralized, polynomial-time algorithm (i.e., greedy solution). We also showed that FASTCOVER runs exponentially faster than the previously proposed distributed algorithms. The superior practical performance of FASTCOVER against all the benchmarks was demonstrated through a large set of experiments, including movie recommendation, location recommendation and dominating set (all were implemented with Spark). Our theoretical results combined with the practical performance of FASTCOVER makes it the only existing distributed algorithm for the submodular cover problem that truly scales to massive data.

**Acknowledgment:** This research was supported by Google Faculty Research Award and DARPA Young Faculty Award (D16AP00046).

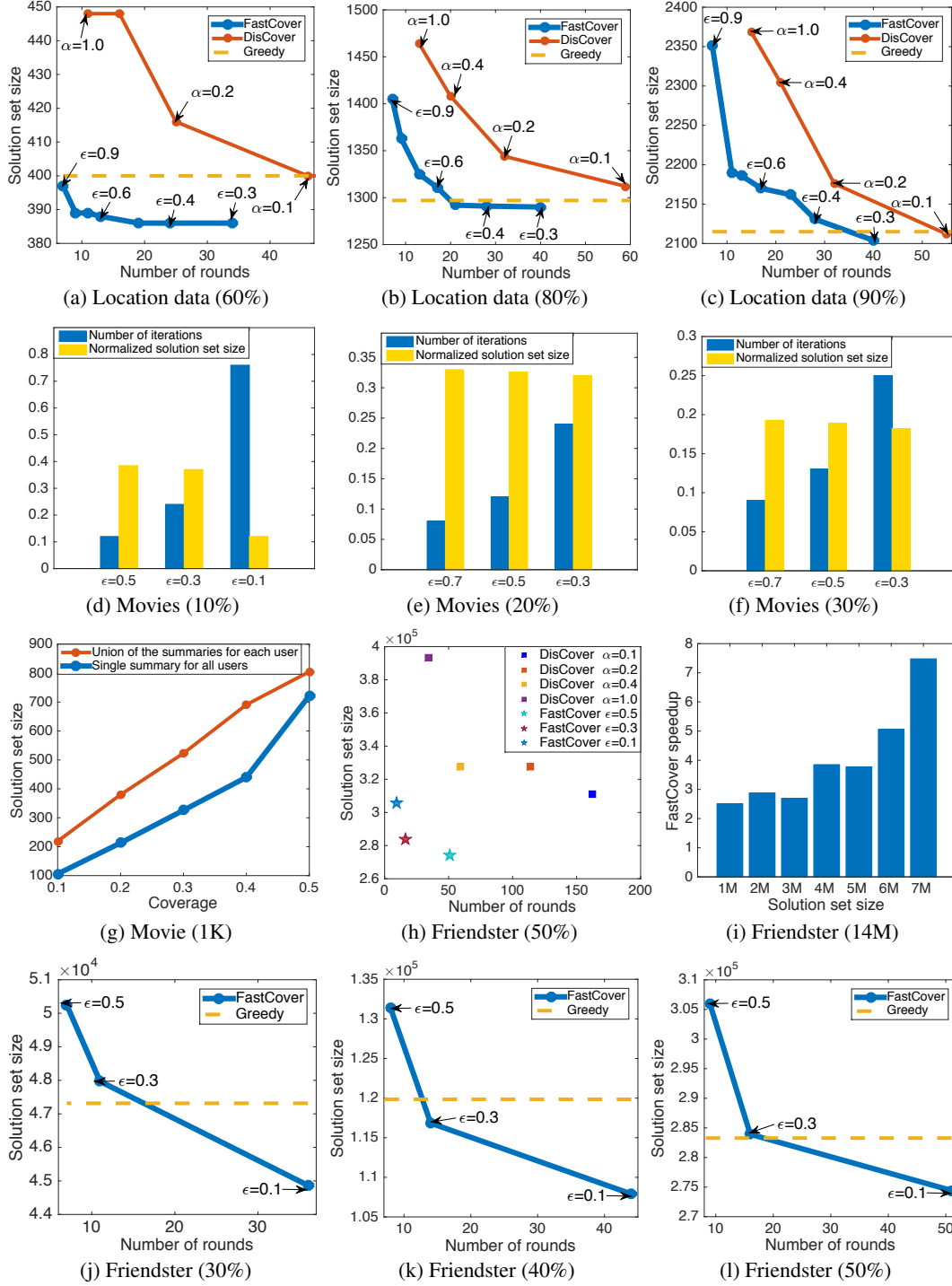

Figure 1: Performance of FASTCOVER vs. other baselines. a), b), c) solution set size vs. number of rounds for personalized location recommendation on a set of 3,056 GPS measurements, for covering 60%, 80%, 90% of the maximum utility of each user. d), e), f) same measures for personalized movie recommendation on a set of 1000 movies, 138,493 users and 20,000,263 ratings, for covering 10%, 20%, 30% of the maximum utility of each user. g) solution set size vs. coverage for simultaneously covering all users vs. covering users one by one and taking the union. The recommendation is on a set of 1000 movies for 1000 users. h) solution set size vs. the number of rounds for FASTCOVER and DISCOVER for covering 50% of the Friendster network with 65,608,366 vertices. i) Exponential speedup of FASTCOVER over DISCOVER on a subgraph of 14M nodes. j), k), l) solution set size vs. the number of rounds for covering 30%, 40%, 50% of the Friendster network.

## Footnotes

[1]All the results are applicable to submodular cover as a special case where there is only public data.

[2]For the submodular cover problem it is a standard assumption that the function is integer-values for the theoretical results to hold. In applications where this assumption is not satisfied, either we can appropriately discretize and rescale the function, or instead of achieving the desired utility $Q$, try to reach $(1 - \delta)Q$, for some $0 < \delta < 1$. In the latter case, we can simply replace $Q$ with $Q/\delta$ in the theorems to get the correct bounds.

[3]Two private lists may point to similar movies, but for now we treat the items on each list as unique entities.

[4]Note that $\sqrt{\min(m, |OPT|)}$ can be as large as $n^{1/6}$ when $|OPT| = n^{1/3}$ and the memory limit of each machine is $n^{2/3}$ which results in $m \geq n^{1/3}$.

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
