[Supplementary Material · fast_cover_full.pdf]

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

# A  Omitted Proofs

Before proving Theorem 4.1, we need to prove a few Lemmas to upper bound the size of final solution and the number of rounds separately.

**Lemma A.1.** FASTCOVER *returns a solution $S$ with at most $|\text{OPT}|\ln(L)/(1-\epsilon)$ items and $f(S) \geq L$.*

*Proof.* We remind that an epoch ends when the if condition of line 14 holds, and therefore we update $\tau$. We prove that at this point all items with marginal value at least $\tau$ have been added to $S$. Therefore the marginal value of every item to $S$ is less than $\tau$ at the end of an epoch. Since $Full_i = False$ for every $i$, all items with marginal value at least $\tau$ are in selected sets $\{S_i\}_{i=1}^m$. In lines $10-12$, FASTCOVER makes sure that every item in $\{S_i\}_{i=1}^m$ with marginal value at least $\tau$ is added to $S$. So at the end of each epoch all marginal values are less than $\tau$. We should also note that by submodularity the marginal values to set $S$ can only decrease, and $\tau$ is unchanged during an epoch, therefore all marginal values are still less than $\tau$ at the end of the epoch.

We now prove that $f(S) \geq L$. The algorithm terminates either at the Break operation of lines 13 in which $f(S) \geq L$ or line 16. At line 16, we are at the end of an epoch, and the else condition holds only if $\tau = 1$. Since all marginal values are less than $\tau = 1$ in this stage, and $f$ only takes integer values, we conclude that all marginal values should be equal to zero. Using submodularity, we conclude that $f(S)$ in this case is equal to $f(\mathbb{V}) \geq L$ because $f(\mathbb{V}) - f(S) \leq \sum_{x \in \mathbb{V}} \Delta(x|S) = 0$ where $\Delta(x|S)$ is $f(S \cup \{x\}) - f(S)$.

We are ready to upper bound $|S|$. Since every time we update $\tau$, it is at least $1-\epsilon$ times its old value, and at the end of each epoch $\tau$ is greater than maximum marginal value to $S$, we can say that throughout the entire algorithm (not just the end of epochs), $\tau$ is always at least $(1-\epsilon)\max_{x \in \mathbb{V}} \Delta(x|S)$. This is in particular true at the beginning of the algorithm that we set $\tau = \max_{x \in \mathbb{V}} f(\{x\})$. Using submodularity of $f$, we know that $\sum_{x \in \text{OPT}} \Delta(x|S) \geq f(\text{OPT}) - f(S)$. So $\max_{x \in \mathbb{V}} \Delta(x|S)$ should be at least $(f(\text{OPT}) - f(S))/|\text{OPT}|$. Since every item we add has marginal value at least $\tau \geq (1-\epsilon)\max_{x \in \mathbb{V}} \Delta(x|S)$, we conclude that each item adds at least a value of $\frac{(1-\epsilon)(f(\text{OPT})-f(S))}{|\text{OPT}|}$. After adding $t$ items, the gap $f(\text{OPT}) - f(S)$ becomes at most $(f(\text{OPT}) - f(\emptyset))(1 - \frac{1-\epsilon}{|\text{OPT}|})^t$. With $t = |\text{OPT}|\ln(L)/(1-\epsilon)$, this gap becomes less than 1, and since $f$ is integral, $f(S)$ should be at least $f(\text{OPT}) = L$ with $|S| = |\text{OPT}|\ln(L)/(1-\epsilon)$ items. $\square$

To upper bound the number of rounds, we categorize all rounds into two groups. We say a round is *good* if the algorithm adds at least $\frac{k}{2}$ items to $S$. Otherwise we call it a *bad* round. We upper bound the number of good and bad rounds separately to reach a unified bound on the total number of rounds of FASTCOVER.

**Lemma A.2.** *The number of good rounds in all epochs is at most $\log_2 L$.*

*Proof.* In a good round, at least $k/2$ items are added to $S$, and each addition increases the value of $f(S)$ by $\tau$. So in a good round, $f(S)$ is increased by at least $k\tau/2$. On the other hand, we define $k$ to be $\lceil (L - f(S^{before}))/\tau \rceil$ where $S^{before}$ is set $S$ just before starting this round. So $f(S)$ is increased by at least $L - f(S^{before})/2$ in this good round. In other words, the difference $L - f(S)$ is reduced by at least a multiplicative factor of 2 in each good round. Once this difference goes below 1, we know $f(S) \geq L$, and the algorithm terminates. Therefore there are at most $\log_2 L$ good rounds in total. $\square$

Next we bound the total number of bad rounds. Since in each epoch, we reduce $\tau$ by a factor of $(1-\epsilon)$ until it becomes at most 1, the number of epochs is upper bounded by $1 + \log_{\frac{1}{1-\epsilon}}(M) \leq \frac{\log(M)}{\log(1/(1-\epsilon))} \leq \frac{\log(M)}{\epsilon}$. Therefore we need to upper bound the number of bad rounds in each epoch.

**Lemma A.3.** *The number of bad rounds in each epoch is at most $\log_{3/2}(n/km)$ with high probability.*

*Proof.* In an epoch, the value of $\tau$ is unchanged, and we keep adding items to $S$. So the set of items that each machine could potentially send to the central machine (items with marginal value at least $\tau$ to set $S$) only shrinks. We call these items candidate items. At the beginning of an epoch, there are at

most $n/m$ such candidate items in each machine since $T_i$ has $n/m$ items. The epoch ends when each machine has at most $k$ candidate items. We show that in each bad round, this set of candidate items shrinks by at least a factor of $2/3$ with high probability, and therefore the number of bad rounds in each epoch is no more than $\log_{3/2}(n/km)$.

Now we focus on a bad round, and how it changes the set of candidate items in a machine $i$. Let $S^{before}$ and $S^{after}$ be the values of set $S$ before and after a bad round. We note that $S^{after} \setminus S^{before}$ has less than $k/2$ items. We define $S_i^{before}$ to be $\{x|x \in T_i \ \& \ f(S^{before} \cup \{x\}) - f(S^{before}) \geq \tau\}$ which is the set of candidate items of machine $i$ before this round. We note that set $S_i$ is a random subset of $S_i^{before}$ with size at most $k$.

We similarly define $S_i^{after}$ to be $\{x|x \in T_i \ \& \ f(S^{after} \cup \{x\}) - f(S^{after}) \geq \tau\}$ which is the set of candidate items in machine $i$ in the next round. We prove that with high probability the size of $|S_i^{after}| \leq 2|S_i^{before}|/3$.

If there are at most $k$ items in $S_i^{before}$, the whole set $S_i^{before}$ is sent to the central machine, and each item in it is either added to $S$ or its marginal value to $S$ becomes less than $\tau$ after this round. So $S_i^{after}$ is empty in this case. In the other case, $k$ random items in $S_i^{before}$ are selected to be sent to the central machine. For the sake of analysis, we define an intermediary hypothetical set $S_i^{hyp}$ which is a set sandwiched between $S_i^{after}$ and $S_i^{before}$. Let $S_i^{hyp}$ be the set $\{x|x \in S_i^{before} \ AND \ f(S^{after} \cup \{x\}) - f(S^{after} \setminus \{x\}) \geq \tau\}$. This is the set of items that either they are candidate items in the next round (part of $S_i^{after}$) or they were added to $S$, and if we remove them from $S$, they become a candidate item in the next round. By definition, we have $S_i^{after} \subseteq S_i^{hyp} \subseteq S_i^{before}$. The significance of $S_i^{hyp}$ is that any item machine $i$ chooses from it to send to the central machine will be chosen by definition. So if $S_i^{hyp}$ has $p$ fraction of $S_i^{before}$ for some $0 \leq p \leq 1$, in expectation $pk$ items out of $k$ items of $S_i$ will be selected in this round. Using concentration bounds, we know that if $\frac{|S_i^{hyp}|}{S_i^{before}}$ is at least $\frac{2}{3}$, with high probability at least $k/2$ selected items in $S_i$ are in $S_i^{hyp}$, and consequently will be added to $S$. However we know that we are in a bad round, and less than $k/2$ items are added to $S$. Therefore with high probability $|S_i^{hyp}|$ is less than $2|S_i^{before}|/3$, and consequently $|S_i^{after}|$ is also less than $2|S_i^{before}|/3$ which completes the proof. $\qquad\square$

Next we summarize all lemmas and prove our main guarantees for the number of rounds of Algorithm FASTCOVER.

**Proof of Theorem 4.1** Lemma A.1 provides the desired upper bound on size of the solution. Using Lemmas A.2, and A.3, we know that there are at most $\log_2(L)$ good rounds in total, and $\log_{3/2}(n/km)$ bad rounds in each epoch with high probability. We have also proved that there are not more than $\log(M)/\epsilon$ epochs. Therefore the total number of rounds is upper bounded by $\log_{3/2}(n/km)\log(M)/\epsilon + \log_2(L)$. $\qquad\square$

**Remark A.4.** *Each machine sends back at most $k$ items. In proof of Theorem 4.1, we showed that $\tau$ is always at least $(1-\epsilon)$ times the maximum marginal value to set $S$. Using submodularity, we know that $L - f(S) \leq f(\text{OPT}) - f(S) \leq \sum_{x \in \text{OPT}} \Delta(x|S)$. So $k = (L - f(S))/\tau$ cannot be more than $|\text{OPT}|/(1-\epsilon)$. The space requirement for the central machine is $km \leq m|\text{OPT}|/(1-\epsilon)$, and for each distributed machine is $n/m$. Therefore our overall space requirement is no more than $\max\{n/m, m|\text{OPT}|/(1-\epsilon)\}$.*