[Reviews · NeurIPS 2016]

Reviewer 1

Summary

The paper considers the problem of summarizing user data that has public components, together with portions that are private to each user. The goal is to come up with a single summary, such that the "utility" for every user exceeds a given threshold, and the objective is to minimize the size of this summary. The paper observes that this reduces directly to the problem of finding a subset S of a universe V so as to minimize |S|, while ensuring f(S) >= threshold, for a single submodular function f (which, in the application, will be determined by the private data of the users). The paper then presents a greedy algorithm for this problem, which can also be implemented in a distributed way, using polylog(|V|)/\epsilon "rounds" of MapReduce/Hadoop. The algorithm is then implemented on three data sets, where a comparison of quality/speed is made with an earlier algorithm DisCover. (And in one dataset, with the single machine Greedy algorithm.)

Qualitative Assessment

The motivation of public/private data summarization is not captured too well by the problem statement. In particular, finding _one set_ S, which gives a good value for all the users, restricted to what is visible to them, seems far fetched. In practical settings, one may want very different summaries for users. Moreover, minimizing just the _size_ of the summary seems like a very restricted objective. My other issue is that for a system with a large number of users, the function 'f' could be quite time-consuming to compute (plus it has a long description). This seems very unrealistic to be used in a distributed setting, where all the machines must know the private elements of all the users just to compute 'f'. Setting these issue aside, the algorithm itself is along the lines of known work in this area. For instance, the paper of Kumar, Moseley, Vassilvitskii, Vattani, titled "Fast Greedy Algorithms in MapReduce and Streaming" (which the current paper should cite), uses a very similar technique -- accumulate all candidates that give a gain of at least a certain threshold, and add them sequentially, followed by reducing the threshold. There are some small novelties (e.g., the way the algorithm handles the case of 'too many candidates'), but I think these are minor.

Confidence in this Review

2-Confident (read it all; understood it all reasonably well)


Reviewer 2

Summary

A distributed submodular cover algorithm is presented that can be implemented in a small number of rounds of MapReduce and implemented on Scala. This substantially improves the previous work on this topic [13] whose distributed algorithm has a very poor performance on the number of rounds.

Qualitative Assessment

The main motivation for considering submodular cover in a distributed setting comes from applications in public-private data summarization. Some questions regarding this motivation are listed in the more detailed comments section. Overall, I liked this paper as it addresses a natural optimization problem in the distributed setting. My only concern is that the algorithms and the proof are fairly straightforward. However, the improvement over the previous work and the fact that the overall bounds look fairly tight seem to be strong enough reasons to argue for acceptance. I also enjoyed the fact that the paper provides an implementation and evaluation of the algorithms in Spark. Comments: To my taste, the public-private motivation mentioned in the paper is a bit of a stretch. Line 110: sequential picks => sequentially picks I have multiple questions about the connection between the theoretical formulation of the problem and its motivation. Section 3.1. The authors write: “ The objective is to find the smallest set S of movies V to recommend to all users such that all of them reach a certain utility.” I would suggest to change the wording to make it clear that the personal recommendations will only be selected as S \cap (V^P \cup V_u). Section 3.2: Can the authors give references to literature where these specific modeling assumptions have been used? Section 3.3: I didn’t understand why the authors talk about the “vertex cover” problem while the problem they are describing looks like the “dominating set” problem: https://en.wikipedia.org/wiki/Dominating_set I also didn’t understand how the public-private aspect of the problem is present in this scenario as all the data is public.

Confidence in this Review

3-Expert (read the paper in detail, know the area, quite certain of my opinion)


Reviewer 3

Summary

The paper considers the submodular cover problem in the MapReduce setting. In this problem, given a submodular function f(S), the goal is to find a set S of size as small as possible while satisfying the constraint that the function value f(S) is at least a given value Q. In the previous work, an algorithm named DISCOVER is proposed for this problem but it needed sqrt(m, |OPT|) rounds, where m is the number of machines and OPT is the optimal solution. In this work, a new algorithm is proposed that needs only a polylogarithmic number of rounds. The basic idea is to simulate the descending threshold algorithm in the centralized setting. The elements are distributed on m machines. Given the same threshold and the current solution, all machines return elements whose marginal gains exceed the threshold. If the number of such element is too large, each machine only returns a small random subset of the ones it finds. The paper also gives 3 motivating examples of submodular cover with multiple simultaneous constraints, which can be reduced to the above case via known reduction.

Qualitative Assessment

The new algorithm significantly improves over the previous work [13] in both theoretical bounds and experimental performance. The motivation seems strange to me as it is optimizing the size of the union of the summaries of the users. It would not be surprising that this size is much larger than the size of the summary each user might see and the two sizes might not even be correlated. The privacy motivation confuses me as it seems the reduction from multiple functions to the single function setting is known (the paper cites [15,16] for this reduction). It would be much better to position the paper as an improved algorithm for distributed submodular cover as it is the main contribution of the paper. The model considered in this paper is centralized computation in the sense that the system has access to all relevant information (no private information is withheld from the system) and no formalization of privacy is considered so the discussion regarding privacy is rather distracting from the main message (e.g. the solution presented to an user is affected by the present of other users due to the joint objective, how much privacy is lost?). After the rebuttal, I am still very confused about the privacy consideration. The authors claim in the rebuttal that the presence or absence of user B does not affect summary for user A but I think this is not true. Consider a simple scenario where the ground set is simply {1,2}, both 1 and 2 are public. The utility functions are both modular. For user A, it is f_A({1}) = 2, f_A({2}} = 1. For user B, it is f_B({1}) = 0 and f_B({2}) = 1. The target utility is Q=1. If the system only has user A, it can return {1}. However, if the system supports both A and B, it will return {2}. Thus, user A can detect the presence of user B based on the his summary.

Confidence in this Review

2-Confident (read it all; understood it all reasonably well)


Reviewer 4

Summary

This work introuduces the public-private data summarization framework. And they propose an efficient distributed algorithm: FastCover, which outperforms perviously proposed algorithm both theoretically and emperically.

Qualitative Assessment

This work proposed a new distributed cover problem that significantly improves compared to the previous DisCover algorithm. Bellow is the questions& comments: - The FastCover is able to solve the general submodular cover problem, so does the DisCover. Is the superiority of FastCover over DisCover depends on the specific public-private structure of the problem? Will the superiority be preserved for some other submodular cover problem?? For general other submodular cover problem, it has been verified for vertex cover problem, however, it would be better to analyze how the structure of public-private problem, e.g., the fraction of public, private data, influence the performance. - As a distributed algorithm, it is better to analyze the communication cost. --Minor errors: -line 179 $f: 2^V -> R$ should be $f: 2^V -> Z$ - line 181 the marginal notation is not consistent with the convention in line 92

Confidence in this Review

2-Confident (read it all; understood it all reasonably well)


Reviewer 5

Summary

This paper is an extension of reference [13], which summarizes the data with monotone submodularity in a patermeterized way. Here in this paper, the authors introduce the privacy constraints application(s) to data summarization. The proposed FastCover algorithm shows better result with smaller summary set satisfying the criteria in fewer search rounds, compared to the existing DisCover algorithm [13] and the centralized greedy search.

Qualitative Assessment

Some suggestions below: - Discussion on some extreme cases to better evaluate the performance: small V_u, large V_u, large V^P - In lines #151-#152, "The objective is to find the smallest set S of movies V to recommend to all users such that all of them reach a certain utility" does not mean "personalized movie recommendation" (line #134) at all. Please correct me if I am wrong. In personalized movie recommendation, we recommend different movies to different users according to their own available high-ranked movies. The same concern applies to 3.2 Personalized Location Recommendation. In fact, the genre-centric proxies in the Fantom paper (see above) makes better sense for providing privacy+personalization+recommendation under the hood of "genre". - Please provide more result on changing the parameters of FastCover. Like the Figure 2 (a) and (b) of [13]. This helps better compare FastCover with DisCover under different scenarios. - In lines #120-#125, provide real DisCover example(s) for increasing m (number of machines) resulting in increase in number of rounds. This helps the readers to see the real shortcoming of the existing methods. - How does alpha_u impact the location recommendation result in Figure 1(a)-(c)? - Provide explicit description about how f_u is normalized (lines #93-#94). And how sensitive is the normalized f_u to small V_u? For example, assuming alpha is 1 in Eq.3, how different between the following two cases will f_u increase when adding an element to S? a)V_u only contains one ranked movie; b)V_u is V^P

Confidence in this Review

2-Confident (read it all; understood it all reasonably well)


Reviewer 6

Summary

Recommender systems feed the users' data to their algorithms in order to recommend the personalized results to the users. This data can be categorized as public or private for each user. The goal is to find a small and representative summary of this data which is customized for each user. This summarization problem can be viewed as a coverage problem with a submodular scoring function over the whole dataset while maintaining the privacy of the data. From this viewpoint, a simple greedy algorithm is not fast enough to be run over the whole dataset for each user separately. Thus we can aggregate all of their scoring functions in a single submodular function $f$ which cares for all of the users simultaneously. By doing this trick, the data can be distributed among different computing nodes and get summarized in a distributed framework. They prove that this formulation achieves the mentioned requirements by finding an upper bound on the number of the needed MapReduce rounds and the size of the summarized output set.

Qualitative Assessment

Although aggregating (augmenting) all the users' scoring functions into a single submodular function seems to be a good trick, it has not been mentioned how hard is to compute the value of this function in practice. This is an important missing part of this method.

Confidence in this Review

3-Expert (read the paper in detail, know the area, quite certain of my opinion)